# Daily Caffeine Consumption May Increase the Risk of Acute Kidney Injury Related to Platinum-Salt Chemotherapy in Thoracic Cancer Patients: A Translational Study

**DOI:** 10.3390/nu16060889

**Published:** 2024-03-19

**Authors:** Aghiles Hamroun, Antoine Decaestecker, Romain Larrue, Sandy Fellah, David Blum, Cynthia Van der Hauwaert, Arnaud Scherpereel, Alexis Cortot, Rémi Lenain, Mehdi Maanaoui, Nicolas Pottier, Christelle Cauffiez, François Glowacki

**Affiliations:** 1Public Health-Epidemiology Department, Nephrology Department, Lille University Hospital Center, 59000 Lille, France; 2Lille University, UMR1167 RID-AGE, Institut Pasteur de Lille, Inserm, Lille University Hospital Center, 59000 Lille, France; 3Nephrology Department, Alexandra Lepève Hospital—Dunkerque Hospital Center, 59240 Dunkerque, France; 4Lille University, CNRS, Inserm, Lille University Hospital Center, Institut Pasteur de Lille, UMR9020-U1277—CANTHER—Cancer Heterogeneity, Plasticity and Resistance to Therapies, 59000 Lille, France; 5Toxicology and Genetic Diseases Department, Lille University Hospital Center, 59000 Lille, France; 6Lille University, Inserm, Lille University Hospital Center, UMR-S1172 LilNCog, Lille Neuroscience and Cognition, 59000 Lille, France; 7Alzheimer and Tauopathies, LabEx DISTALZ, 59045 Lille, France; 8Thoracic Oncology Department, Lille University Hospital Center, 59000 Lille, France; 9Nephrology Department, Lille University Hospital Center, 59000 Lille, France; 10Lille University, U1190 Translational Research for Diabetes, Inserm, Institut Pasteur de Lille, 59000 Lille, France

**Keywords:** cisplatin, carboplatin, acute kidney injury, AKI, nephrotoxicity, chemotherapy, caffeine, thoracic cancer

## Abstract

Although their efficacy has been well-established in Oncology, the use of platinum salts remains limited due to the occurrence of acute kidney injury (AKI). Caffeine has been suggested as a potential pathophysiological actor of platinum-salt-induced AKI, through its hemodynamic effects. This work aims to study the association between caffeine consumption and the risk of platinum-salt-induced AKI, based on both clinical and experimental data. The clinical study involved a single-center prospective cohort study including all consecutive thoracic cancer patients receiving a first-line platinum-salt (cisplatin or carboplatin) chemotherapy between January 2017 and December 2018. The association between daily caffeine consumption (assessed by a validated auto-questionnaire) and the risk of platinum-salt induced AKI or death was estimated by cause-specific Cox proportional hazards models adjusted for several known confounders. Cellular viability, relative renal NGAL expression and/or BUN levels were assessed in models of renal tubular cells and mice co-exposed to cisplatin and increasing doses of caffeine. Overall, 108 patients were included (mean age 61.7 years, 65% men, 80% tobacco users), among whom 34 (31.5%) experienced a platinum-salt-induced AKI (67% Grade 1) over a 6-month median follow-up. The group of high-caffeine consumption (≥386 mg/day) had a two-fold higher hazard of AKI (adjusted HR [95% CI], 2.19 [1.05; 4.57]), without any significant association with mortality. These results are consistent with experimental data confirming enhanced cisplatin-related nephrotoxicity in the presence of increasing doses of caffeine, in both in vitro and in vivo models. Overall, this study suggests a potentially deleterious effect of high doses of daily caffeine consumption on the risk of platinum-salt-related AKI, in both clinical and experimental settings.

## 1. Introduction

Up to 50 to 70% of cancer patients undergoing chemotherapy receive at least one course of platinum-salt-based chemotherapy (mainly cisplatin, carboplatin and oxaliplatin), either alone or in combination, as adjuvant or neoadjuvant treatment for various oncological indications such as lung, upper aerodigestive tract, uterine, ovarian or bladder neoplasms [1,2,3].

The use of cisplatin is limited by the frequent occurrence of serious adverse events, primarily nephrotoxicity [4]. On average, this toxicity affects 30 to 40% of patients after multiple courses, and up to 30% of patients exposed to a single dose of cisplatin [2,4]. This adverse renal event thus affects the patients’ prognosis through two mechanisms: acute kidney injury (AKI), which is associated with higher morbidity and mortality, and the risk of reducing or even discontinuing platinum-based chemotherapy [4,5,6]. The risk factors are well known, including baseline kidney function, general condition, chemotherapy dose and the frequency of administration [4,6,7]. The pathophysiology is also well-described, involving multifactorial mechanisms such as hemodynamic changes, inflammation, oxidative stress, apoptosis and fibrogenesis [4,8]. These mechanisms are reflected downstream by the elevation of serum creatinine and blood urea nitrogen (BUN), indicative of significant kidney damage. Additional data suggest that certain urinary biomarkers appear earlier during kidney impairment, especially in the case of cisplatin-induced nephrotoxicity, such as neutrophil gelatinase-associated lipocalin (NGAL), and thus constitute relevant outcomes for preclinical studies [9].

Numerous studies have investigated the prevention of platinum-salt-induced renal toxicity, involving various agents, with often inconclusive results [8,10,11,12]. Currently, there is no consensus on a preventive method, except for intravenous hydration and magnesium supplementation [13]. Among the studied agents, adenosine and its receptors appear as potential targets of interest, notably due to their effects on renal hemodynamics [14,15,16].

Coffee, the second most consumed beverage in the world (around 200 mg per day in Europe) [17,18], has been found to be a natural non-selective inhibitor of adenosine A_1_, A_2A_, A_2B_ and A_3_ receptors [19,20,21,22,23]. A significant increase in diuresis is observed with a few cups of coffee, this effect being mediated by A_1_ receptors. Through its antagonistic action on the A_1_ receptor, caffeine also inhibits the tubulo-glomerular reflex and thus influences renal hemodynamics. While there are literature data supporting the beneficial effects of caffeine consumption on kidney health outcomes [24,25], such as a potentially decreased incidence of chronic kidney disease [26,27,28], there are currently no data regarding the risk of platinum-salt-induced AKI.

The principal aim of our study was to investigate the association between daily caffeine consumption and the occurrence of platinum-salt-induced AKI within a population of patients with lung cancer. The secondary objective was to investigate this association experimentally based on both in vitro (renal tubular cells) and in vivo models (murine models) of joint exposure to cisplatin and caffeine.

## 2. Methods

### 2.1. Clinical Study

The present work is reported in accordance with the Strengthening the Reporting of Observational Studies in Epidemiology (STROBE) guidelines [29].

***Study setting and population.*** We conducted a single-center prospective cohort study which included all consecutive adult patients with a diagnosed thoracic cancer (including lung adenocarcinoma, epidermoid carcinoma, and mesothelioma) receiving a first-line platinum-salt chemotherapy between January 2017 and December 2018 at the Lille University Hospital Center. All patients provided informed consent to participate in the study, which was approved by the Institutional Review Board (Registration Number: #DEC16-265).

***Data collection.*** For each patient, we collected various data from electronic medical records, including demographics (age, sex, body mass index), medical history (diabetes, hypertension, hearth failure, cardiovascular disease), smoking status, the stage and histological type of neoplasia, baseline laboratory data such as serum creatinine (and estimated glomerular filtration rate—GFR—according to the CKD-EPI formula), albumin, pre-albumin and C-reactive protein levels; information on the type of platinum salt, dosage, number of cycles and co-administered chemotherapy molecule; and other daily treatments.

***Caffeine consumption assessment.*** Usual caffeine intake over the last ten years was assessed by a validated self-administered questionnaire on Day 0 of the first platinum-salt-based chemotherapy cycle. The mean daily intake of caffeine containing items (coffee, tea, chocolate, sodas) was reported, along with any changes in consumption over the 10-year period. The mean caffeine intake before chemotherapy initiation was calculated using standardized conversion tables (Appendix A). The reproducibility of the questionnaire was previously validated, providing an accurate measurement of daily consumption in mg/day while supporting relatively consistent consumption over time [30].

***Outcomes.*** The occurrence and staging of acute kidney injury (AKI) were defined according to the KDIGO 2012 guidelines [31]. Chemotherapy-related toxicity events were defined in accordance with the Common Terminology Criteria for Adverse Events (CTCAE), v5.0 [32]. All outcomes, including deaths, were identified from patients’ medical records. For time-to-event analyses, we focused on the time until the occurrence of a first AKI episode and the follow-up period was defined as the time between the first administration of platinum-based chemotherapy and the date of the first event among AKI, change in chemotherapy regimen (platinum salt withdrawal), death or last follow-up (endpoint date on 31 December 2019).

***Statistical analyses.*** Descriptive data were reported according to their nature and distribution: in counts and percentages for categorical variables, as the median (1st; 3rd quartile) for non-normally distributed quantitative data, and as the mean (±standard deviation) for normally distributed quantitative data. The normality of continuous variables was assessed graphically and using the Shapiro–Wilk test. Given the violation of log-linearity assumptions, two patient groups were identified according to the median caffeine consumption (i.e., 386 mg per day within the study population, see Figure 1): “high caffeine consumption” and “moderate caffeine consumption”, which are consistent with the threshold identified in the literature for excessive consumption [33]. Similarly, two categories of chemotherapy dose, “high dose” and “low dose”, were based on the medians of the prescribed platinum-based chemotherapy dose during the first cycle. Bivariate comparisons between the high and moderate caffeine consumption groups were performed as follows: Student’s *t*-test for mean comparisons, Chi-square or Fisher’s exact test for proportion comparisons, and the Wilcoxon test for distribution comparisons. Cumulative incidences of platinum-salt-related AKI and death, accounting for competing risks, were estimated using the Aalen–Johansen method, and further compared according to the caffeine consumption group using the Gray test. The hazard ratios (95% confidence interval) of AKI and death associated with the higher daily caffeine consumption group were estimated using multivariable cause-specific Cox proportional hazards models. These analyses considered relevant confounding factors from the literature and used a backward selection of the outcome-associated variables in univariate analyses (*p* < 0.10) to minimize the Bayesian information criterion. The proportional hazards assumption was assessed graphically and using the Schönfeld residuals test. Potential interactions between the exposure (caffeine consumption) and the confounders were also tested. The threshold for statistical significance for all tests was set at 5%. The analyses were conducted using R software, version 3.6.2.

### 2.2. In Vitro Model

***Cell culture.*** RPTECs (renal proximal tubular epithelial cells) immortalized with a pLXSN-hTERT1 retroviral vector (CRL-4031, ATCC) are a relevant in vitro model to evaluate cisplatin’s deleterious effects [34,35,36]. Cells were cultured in DMEM-F12 medium (Thermo Fisher Scientific, Waltham, MA, USA) supplemented with 1% penicillin–streptomycin, 5 pmol/L triiodo-l-thyronine, 10 ng/mL recombinant human EGF, 3.5 μg/mL ascorbic acid, 5.0 μg/mL human transferrin, 5.0 μg/mL insulin, 25 ng/mL prostaglandin E1, 25 ng/mL hydrocortisone, 8.65 ng/mL sodium selenite, 0.1 mg/mL G418, and 1.2 g/L sodium bicarbonate (Sigma Aldrich, St. Louis, MO, USA). Cells were cultured at 37 °C in a humidified atmosphere of 5% CO_2_.

***In vitro cytotoxicity assay.*** Twenty-four hours after seeding in 96-well plates (20,000 cells/well), RPTEC cells were exposed to 0–25 μM cisplatin (Mylan) and 0–6.6 mM caffeine (Sigma Aldrich) for 48 h. Cell viability was evaluated using the CellTiterGlo kit (Promega, Madison, WI, USA), based on ATP quantitation, according to the supplier’s recommendations. Cells exposed to medium only were used as the reference (100% viability).

### 2.3. Mouse Model of Cisplatin-Induced Nephrotoxicity

All animal care and experimental protocols were approved by the Institutional Animal Care and Use Committee (IACUC) of Lille University (Protocol Number: 2018101215473925). Manipulators carried out all experimental protocols under strict guidelines to ensure careful and consistent handling of the mice. A total of 72 mice were used in this study. Sample size was chosen empirically based on our previous experiences in the calculation of experimental variability; no statistical method was used to predetermine sample size and the number of samples. The experiments in this study were conducted in accordance with the ARRIVE guidelines [37].

Animal procedures were performed in 8 to 10-week old male C57Bl6/J mice (Janvier Labs). After one week of acclimatization, mice were randomly separated and allocated to control or treatment groups. Caffeine (Sigma-Aldrich) was administered in the drinking water (0 g/L, 0.3 g/L or 1 g/L) for three weeks. On average, mice drink 5 mL per day, giving a daily dose of 1.5 mg or 5 mg caffeine to each mouse, depending on the caffeine concentration in the drinking water. These exposure levels were chosen as they are representative of a 500 mg and 1.5 g daily intake in human, respectively. Cisplatin (Accord Healthcare) was dissolved in saline solution. Two weeks after caffeine administration, acute cisplatin nephrotoxicity was induced following a single intra-peritoneal injection of 10 mg/kg cisplatin [38] (n = 17 per caffeine concentration) or vehicle (n = 7). Three days after this single injection, animals were sacrificed by cervical dislocation. Prior to sacrifice, retro-orbital blood samples were collected in heparinized tubes and centrifuged for 10 min at 2000× *g* at room temperature. Kidney function was blindly assessed by blood urea nitrogen (BUN) measurement using an AU480 Chemistry Analyzer (Beckman Coulter, Brea, CA, USA). At the time of sacrifice, kidneys were harvested and stored in RNA later solution (Thermo Fisher). Cotton sticks were placed in the cages to reduce mouse stress during the entire procedure. As no limit point (drastic weight loss) was reached during the study, no sample mice or data points were excluded from the reported analyses.

### 2.4. Gene Expression

***RNA extraction.*** Total RNA from cultured cells was extracted using the RNeasy Mini kit (Qiagen, Hilden, Germany) following the manufacturer’s recommendations. Regarding kidney specimen, total RNA was extracted with phenol/chloroform and subsequently precipitated in isopropanol as described previously [39].

***Quantitative PCR.*** mRNA retro-transcription was performed using the High-Capacity cDNA reverse transcription kit (Thermo Fisher Scientific). Quantitative PCR was performed on a StepOnePlusTM Real-Time PCR System (Thermo Fisher Scientific) using Universal Master Mix (Thermo Fisher Scientific) and the following TaqMan assays: NGAL (neutrophil gelatinase-associated lipocalin) (assay ID m01324470_m1 for mouse samples and assay ID Hs00194353_m1 for cells) and PPIA (assay ID Mm02342430_m1 for mouse samples and assay ID Hs99999904_m1 for cells). For normalization, transcript levels of PPIA (cyclophilin A) were used as endogenous control for gene expression. Relative expression levels of mRNAs were assessed using the comparative threshold cycle method (2^−ΔΔCT^) [40].

### 2.5. Statistical Analyses

Statistical analyses were performed using GraphPad Prism software, version 10.2.1. The results are given as mean ± SEM. Two-tailed Mann–Whitney test was used for single comparisons; a one-way ANOVA followed by the Bonferroni post hoc test was used for multiple comparisons. A *p*-value of less than 0.05 was considered statistically significant.

## 3. Results

### 3.1. Study Population

A total of 108 patients diagnosed with lung cancer and eligible for first-line platinum-salt chemotherapy were enrolled during the study period. This population was predominantly male, with a mean age of 61.7 years. Half of them had hypertension, while a quarter had cardiovascular history. Most patients (around 80%) were active or former smokers, with an average consumption exceeding 30 pack-years. Regarding the lung neoplasia, the most observed histological subtype was adenocarcinoma (57.4%), predominantly at advanced stages III or IV. The distribution between cisplatin and carboplatin was relatively balanced within the population. Pemetrexed was the most frequently used co-drug, accounting for over half of the cases (Table 1).

In this population, a high daily caffeine consumption was observed, with a median estimated at 386 mg/day (252–658) (Figure 1). The patients were comparable in all the characteristics previously described according to their daily caffeine consumption, except for a higher representation of men in the high caffeine consumption group (75.9% vs. 55.6%, *p* = 0.04), as well as a slightly higher baseline serum prealbumin level (0.25 ± 0.08 g/L vs. 0.21 ± 0.09 g/L, *p* = 0.02) (Table 1).

### 3.2. Platinum-Salt-Related AKI and Mortality

Over a median follow-up of 192 days (102.5; 257.2), at least one AKI episode occurred in 34 patients (31.5%), of which two-thirds were in the cisplatin group, with a median onset of 70.5 days (22.3; 125.25). In parallel, we observed 47 deaths (43.5%), within a median delay of 150 days (70.5; 301). The cumulative incidence of AKI at 6 months was significantly higher in the group more exposed to caffeine (35.2% (34.4–36.0) vs. 18.7% (18.1–19.3), respectively, for the high and moderate caffeine consumption groups, *p* = 0.03) (Figure 2).

After adjusting for confounders, the group of high caffeine consumption had a two-fold higher hazard of platinum-salt-related AKI (adjusted HR (95% CI), 2.19 (1.05; 4.57), *p* = 0.04), without any significant association with death (adjusted HR (95% CI), 1.11 (0.61; 2.03), *p* = 0.74) (Table 2). Other factors associated with the risk of AKI included the high-dose chemotherapy regimen, a performance status ≥ 1 and a lower baseline kidney function. For mortality, only the high-dose chemotherapy regimen was associated with a better prognosis. Moreover, no interaction was identified regarding the risk of AKI, between the caffeine consumption group and the type of platinum salt, chemotherapy dose or smoking status (all *p*-values > 0.10).

### 3.3. Secondary Outcomes

There was no significant difference in terms of treatment response and disease progression between the two exposure groups (Table 3). The main observed chemotherapy-related adverse effects were hematological and gastrointestinal, occurring respectively in 24% and 20% of cases. There was no difference in the incidence of adverse effects during follow-up, except for more frequent digestive toxicity in the high caffeine consumption group (29.6% vs. 11.1%, *p* = 0.03). The causes of death were also globally comparable between the two groups.

### 3.4. High Doses of Caffeine Promote Cisplatin-Induced Nephrotoxicity Both In Vitro and In Vivo

To further characterize the impact of caffeine on kidney function after cisplatin exposure, we first determined the effect of caffeine on cisplatin-induced cell death in RPTEC/hTERT1, an immortalized cell line derived from renal proximal tubular epithelial cells. This cell model was exposed to cisplatin alone (0–25 μM), caffeine alone (0–6.6 mM) or in combination with both cisplatin and caffeine for 72 h, and cell viability was assessed. These results showed that caffeine exacerbates cisplatin-induced cell death, in particular for the highest dose of cisplatin (Figure 3A). The viability of cells exposed to cisplatin (25 μmol/L) decreased by 15% with 0.33 mmol/L of caffeine (87.0 ± 4.0% vs. 74.0 ± 2.0%; *p* < 0.001), and up to 25% with 6.66 mmol/L of caffeine (40.0 ± 3.0% vs. 30.0 ± 1.0%; *p* < 0.01). Similarly, the mRNA levels of NGAL were significantly increased when cells were cotreated with caffeine compared to cisplatin only (4.1 ± 0.3, 7.8 ± 0.9 and 11.5 ± 1.5, respectively, for 1.25, 2.5 and 5 mmol/L of caffeine, *p* < 0.05) (Figure 3B).

Moreover, caffeine effects on cisplatin-induced nephrotoxicity were evaluated in vivo based on a mouse model that develops significant renal lesions after systemic administration of a single dose of cisplatin, as shown by the increased expression of NGAL (31.5 ± 6.0 vs. 1.0 ± 0.2; *p* < 0.001) (Figure 3C). Caffeine at a low dose did not significantly modulate the cisplatin effect. Indeed, compared to mice only treated with cisplatin, those pretreated with caffeine at a low dose (0.3 g/L) for 2 weeks before cisplatin exposure did not exhibit significant changes in either NGAL expression (46.5 ± 6.5 vs. 31.5 ± 6.0; *p* = 0.10) or BUN (1.9 ± 0.7 vs. 1.4 ± 0.3; *p* = 0.48) (Figure 3C,D). By contrast and consistent with in vitro data, co-administration of caffeine at a high dose worsened cisplatin nephrotoxicity: mice pretreated with caffeine at a high dose (1 g/L) had a significantly altered kidney function as exemplified by an increased NGAL expression (79.2 ± 25.0 vs. 31.5 ± 6.0; *p* < 0.001) and BUN level (2.8 ± 0.8 vs. 1.4 ± 0.3; *p* < 0.05) (Figure 3C,D).

## 4. Discussion

Within this prospective cohort of lung cancer patients, daily caffeine consumption was found to be significantly associated with a higher risk of platinum-salt-induced nephrotoxicity. A two-fold increased hazard of platinum-salt-related AKI was observed in the group with high daily caffeine intake, with no apparent effect on mortality risk. Consistent with these results, the synergistic nephrotoxic effect of caffeine was experimentally confirmed in both cellular and mouse models.

One of the peculiarities of our study is the notably high caffeine consumption compared to the general population [41,42]. Indeed, the median caffeine consumption in our population was 386 mg per day, and over 10% of patients consumed more than 1000 mg per day. In the literature, the proportion of individuals taking more than 400 mg per day ranges from 6% to 33% in general population studies, with higher consumption observed in Nordic countries [18,41]. Several characteristics related to the selection of our population may explain this high caffeine consumption, such as the predominance of men, a relatively advanced age, and the high prevalence of smokers [17,18,41]. Other factors associated with excessive caffeine consumption may contribute to the exposure to high doses of caffeine observed within our population, such as poor socioeconomic status and educational level [42,43].

During a median follow-up of approximately 6 months, limited by the patients’ short life expectancy, the proportion of platinum-salt-induced AKI was 31.5%, with two-thirds being Grade 1, which is comparable to literature data [44]. The occurrence of AKI was also twice as common with cisplatin compared to carboplatin, also consistent with previous findings. Other risk factors for platinum-salt-related AKI identified within our study population, such as chemotherapy dose, poor performance status, and lower baseline kidney function, are also classically described in the literature [4,6,7].

Recent literature seems to highlight the health benefits of moderate chronic coffee consumption [24,25]. In the umbrella review by Poole et al., which included over two-hundred meta-analyses, moderate coffee consumption compared to no consumption was associated with a 10% reduction in all-cause mortality and a 19% reduction in cardiovascular mortality [25]. Furthermore, chronic kidney disease has been listed by the authors as one of the ten most favorable health outcomes linked to coffee consumption. While our findings may appear discordant with the existing literature, they likely find their explanation in the magnitude of daily caffeine consumption observed in our population. Indeed, the median caffeine consumption in our study is around 400 mg per day, which aligns with the threshold of toxicity identified in most studies [45,46]. Interestingly, the European Food Safety Authority recommendations set this toxicity threshold at precisely 400 mg per day [33]. This threshold is mainly justified by the fact that associations described in the literature between caffeine consumption and various health outcomes often exhibit a ”U-shaped curve”, with effects diminishing or potentially becoming detrimental beyond a threshold of 5–6 cups of coffee per day, thus corresponding to 300–350 mg of caffeine [24]. A recent study using the UK Biobank data, involving more than 300,000 individuals, perfectly illustrates this association: while the risk of cardiovascular events, compared to the population drinking one to two cups of coffee per day, appeared to be 11% higher in patients not consuming caffeine, the excess risk reached 22% in patients consuming more than six cups of coffee per day [47].

This phenomenon could be explained by the physiological properties of caffeine: in addition to its non-selective antagonistic action on adenosine receptors, this molecule has the unique characteristic of exhibiting differential effects according to the ingested dose [20,21,46]. Indeed, while only a few milligrams of caffeine are sufficient to exert an antagonistic effect on adenosine A1 receptors, concentrations 20 times higher are needed to inhibit phosphodiesterase, and even 100 times higher to increase intracellular calcium levels through activation of the ryanodine receptors. These signaling pathways are probably not significant for common caffeine consumption but may be activated with higher doses and play a detrimental role on kidney function in that context [46]. These hypotheses are also supported by our experimental findings: while the nephrotoxicity of a fixed dose of cisplatin is relatively reproducible for lower caffeine exposures, it appears significantly more pronounced at higher levels of caffeine intake. These results are concordant whether examining the viability of epithelial tubular cells or the expression of biomarkers of kidney injury in a mouse model exposed to cisplatin. Interestingly, we recently reported that chronic administration of Istradefylline, a selective inhibitor of the adenosine receptor A_2A_, limits cisplatin nephrotoxicity irrespective of the dose [14]. Other mechanisms that may contribute to platinum-salt-related AKI in patients with high caffeine consumption include increased secretion of renin and norepinephrine, as well as excessive activation of the renin–angiotensin–aldosterone system. Another simple explanation could also be related to the hypovolemia induced by the exacerbation of the chemotherapy-related digestive toxicity, as observed in our study population.

The main strengths of this study are related to its originality, its mixed design including prospective epidemiological and experimental data, and the standardized assessment of daily caffeine consumption using a validated self-questionnaire. Additionally, it identifies well-established risk factors for platinum-salt-related AKI and demonstrates concordant results across in vitro, in vivo and epidemiological studies, enhancing its consistency. Another strength of this study is its practical relevance to the population of lung cancer patients, who could potentially be prone to excessive caffeine consumption. Our findings might suggest considering the evaluation of caffeine intake as part of dietary assessment and advocating for more moderate consumption in cases of excessive exposure.

However, our study does have several limitations. First, our observational study included a limited sample size. Nonetheless, the comparability of the groups at baseline based on their caffeine consumption levels suggests a potential for minimal unmeasured confounding. Furthermore, half of the patients received carboplatin-based chemotherapy, which is less associated with AKI, thus limiting the number of observed events. The generalizability of these results, however, remains limited due to the single-center nature of the study and the specificities of this population of patients with often advanced lung cancer. This study nevertheless allows our results to be relevant in a real-life context, which is important for most clinicians. Lastly, this study did not assess the effect of moderate caffeine consumption due to the exceptionally high daily caffeine exposure within our population and the limited sample size for complementary subgroup analyses.

## 5. Conclusions

This study is the first, to our knowledge, to investigate the relationship between daily caffeine consumption and the risk of platinum-salt-related AKI with both epidemiological and experimental data. It highlights a doubled hazard of AKI among patients consuming over 400 mg of caffeine per day. These results were also confirmed by enhanced cisplatin-related nephrotoxicity in the presence of high caffeine exposure in both cellular and mouse models. Further studies are needed to evaluate the effect of daily caffeine consumption more precisely, especially at more moderate doses.

## Figures and Tables

**Figure 1 nutrients-16-00889-f001:**
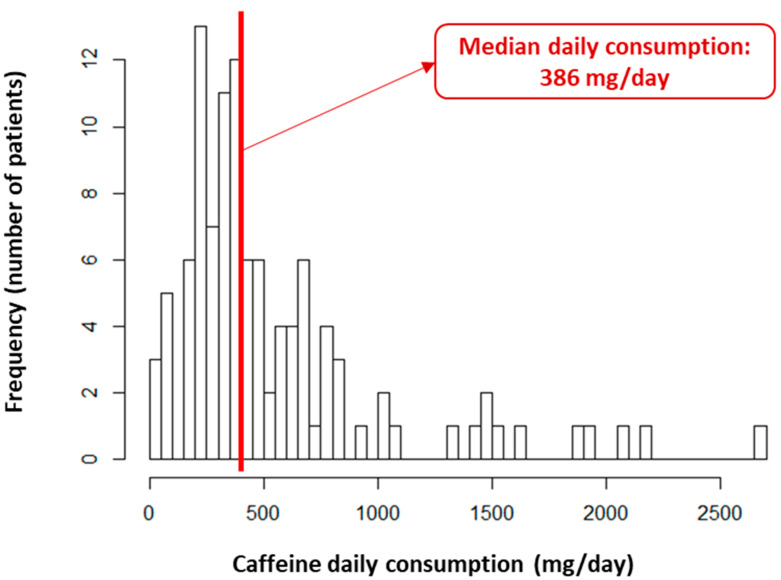
Repartition of the caffeine daily consumption within the study population.

**Figure 2 nutrients-16-00889-f002:**
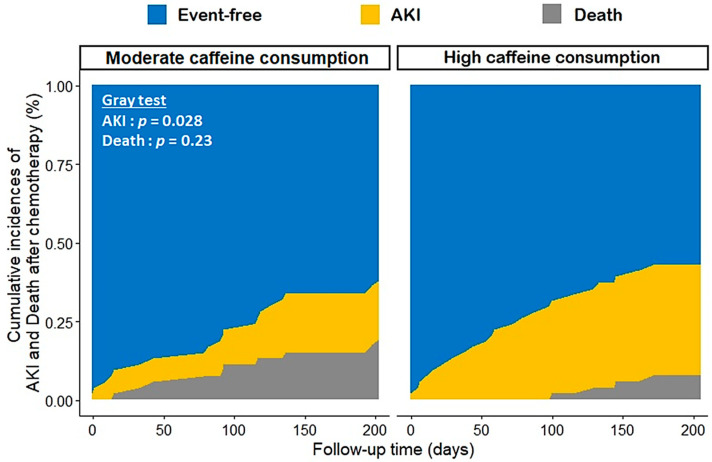
Cumulative incidences of the competitive risks of AKI and death according to the level of daily caffeine consumption.

**Figure 3 nutrients-16-00889-f003:**
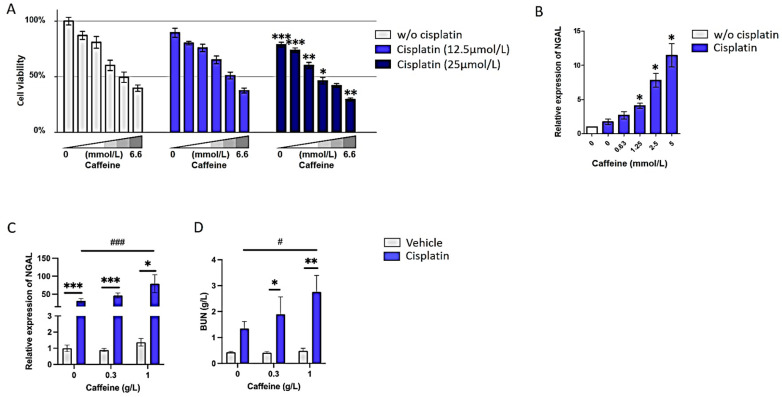
Caffeine affects cisplatin-induced toxicity in a dose-dependent manner. (**A**) Bar charts showing the effects of increasing doses of caffeine on the viability of RPTEC/hTERT cells exposed to cisplatin at a fixed concentration (12.5 μmol/L or 25 μmol/L). (**B**) Bar charts showing the dose-dependent effect of caffeine on NGAL mRNA level measured in RPTEC/hTERT cells exposed to cisplatin at a concentration of 12.5 μmol/L. n = 2–3 independent experiments. Data are shown as the mean +/− SEM. *: *p* < 0.05; **: *p* < 0.01; ***: *p* < 0.001. (**C**,**D**) Bar charts showing the (**C**) NGAL relative renal expression and (**D**) plasmatic level of BUN in mice that received a single dose of cisplatin (10 mg/kg) and exposed to increasing doses of caffeine for two weeks before cisplatin administration and one week after. n  =  6–16 mice per group. Data are shown as the mean +/− SEM. *: *p* < 0.05; **: *p* < 0.01; ***: *p* < 0.001 versus vehicle; #: *p* < 0.05; ### *p* < 0.001 versus cisplatin (10 mg/kg) without caffeine exposure. NGAL: neutrophil gelatinase-associated lipocalin; BUN: blood urea nitrogen.

**Table 1 nutrients-16-00889-t001:** Characteristics of the study population, according to the level of daily caffeine consumption.

	Overall N = 108	Daily Caffeine Consumption (</≥386 mg/Day)		
		Moderate N = 54	High N = 54	*p*-Value
Demographic characteristics				
Age (years)	61.7 (±9.44)	61.4 (±9.96)	62.1 (±8.98)	0.686
Male sex	71 (65.7%)	30 (55.6%)	41 (75.9%)	0.043
Body mass index (kg/m^2^)	24.2 [21.3; 27.0]	23.9 [20.3; 26.9]	24.2 [21.4; 27.0]	0.381
Performance status (≥1)	63 (58.3%)	29 (53.7%)	34 (63.0%)	0.435
Medical history				
Diabetes	15 (13.9%)	8 (14.8%)	7 (13.0%)	1.000
Hypertension	55 (50.9%)	26 (48.1%)	29 (53.7%)	0.700
Heart failure	7 (6.48%)	4 (7.41%)	3 (5.56%)	1.000
Cardiovascular disease	30 (27.8%)	12 (22.2%)	18 (33.3%)	0.283
Tobacco use				0.337
* Never*	22 (20.4%)	14 (25.9%)	8 (14.8%)	
* Former*	49 (45.4%)	22 (40.7%)	27 (50.0%)	
* Active*	37 (34.3%)	18 (33.3%)	19 (35.2%)	
Tobacco quantity (*Pack-Year*)	32.4 (±25.3)	31.2 (±27.2)	33.5 (±23.5)	0.645
Malignancy characteristics				
Histological type				0.395
* Adenocarcinoma*	62 (57.4%)	29 (53.7%)	33 (61.1%)	
* Squamous cell carcinoma*	7 (6.48%)	2 (3.70%)	5 (9.26%)	
* Pleural mesothelioma*	13 (12.0%)	7 (13.0%)	6 (11.1%)	
* Other*	26 (24.1%)	16 (29.6%)	10 (18.5%)	
Stage				0.589
* I*	3 (2.78%)	2 (3.70%)	1 (1.85%)	
* II*	14 (13.0%)	5 (9.26%)	9 (16.7%)	
* III*	17 (15.7%)	11 (20.4%)	6 (11.1%)	
* IV*	57 (52.8%)	28 (51.9%)	29 (53.7%)	
* Unclassified*	17 (15.7%)	8 (14.8%)	9 (16.7%)	
Metastatic status	63 (60.0%)	33 (61.1%)	30 (58.8%)	0.968
Baseline biological findings				
Creatinine (mg/L)	0.77 (±0.23)	0.75 (±0.23)	0.79 (±0.23)	0.272
eGFR (*MDRD*, mL/min/1.73 m^2^)	86.1 [67.2; 109]	88.0 [73.4; 110]	79.1 [65.8; 102]	0.212
Albumin (g/dL)	3.82 (±0.60)	3.83 (±0.62)	3.82 (±0.59)	0.979
C-reactive protein (mg/L)	15.7 [3.23; 45.4]	18.9 [4.18; 66.1]	13.2 [3.23; 38.0]	0.305
Prealbumin (g/L)	0.23 (±0.08)	0.21 (±0.09)	0.25 (±0.08)	0.020
Hemoglobin (g/dL)	12.9 (±1.74)	13.0 (±1.70)	12.8 (±1.78)	0.539
Chemotherapy treatment				
Chemotherapy regimen				0.700
* Cisplatin*	51 (47.2%)	24 (44.4%)	27 (50.0%)	
* Carboplatin*	57 (52.8%	30 (55.6%)	27 (50.0%)	
High-dose chemotherapy	55 (50.9%)	29 (53.7%)	26 (48.1%)	0.700
Co-administered drug				0.410
* Etoposide*	16 (14.8%)	10 (18.5%)	6 (11.1%)	
* Navelbine*	20 (18.5%)	7 (13.0%)	13 (24.1%)	
* Pemetrexed*	57 (52.8%)	29 (53.7%)	28 (51.9%)	
* Taxol*	15 (13.9%)	8 (14.8%)	7 (13.0%)	
Other treatments				
Dopamine antagonist	77 (71.3%)	35 (64.8%)	42 (77.8%)	0.202
Setron	17 (15.9%)	10 (18.5%)	7 (13.2%)	0.626
Aprepitant	96 (89.7%)	48 (88.9%)	48 (90.6%)	1.000
Diuretic	15 (13.9%)	5 (9.26%)	10 (18.5%)	0.266
RAAS inhibitor	39 (36.1%)	17 (31.5%)	22 (40.7%)	0.423
Beta-blocker	22 (20.4%)	13 (24.1%)	9 (16.7%)	0.474
Statin	35 (32.4%)	16 (29.6%)	19 (35.2%)	0.681
Biguanide	5 (4.63%)	3 (5.56%)	2 (3.70%)	1.000
NSAID	2 (1.85%)	1 (1.85%)	1 (1.85%)	1.000
Steroid	104 (96.3%)	52 (96.3%)	52 (96.3%)	1.000
Folate	53 (49.1%)	23 (42.6%)	30 (55.6%)	0.248
Proton pump inhibitors	32 (29.9%)	12 (22.2%)	20 (37.7%)	0.123

eGFR: estimated glomerular filtration rate; RAAS: renin–angiotensin–aldosterone system; NSAID: non-steroidal anti-inflammatory drug.

**Table 2 nutrients-16-00889-t002:** Multivariable cause-specific Cox proportional hazards models: platinum-salt-related AKI and mortality risks associated with high caffeine consumption (N = 108 patients).

	AKI HR [95% CI]	*p*-Value	Death HR [95% CI]	*p*-Value
Daily caffeine consumption				
* moderate* (<386 mg/day)	ref	-	ref	-
* high* (≥386 mg/day)	2.19 (1.05; 4.57)	0.04	1.11 (0.61; 2.03)	0.74
Chemotherapy regimen				
* Cisplatin*	ref	-	-	-
* Carboplatin*	0.50 (0.23; 1.10)	0.11	-	-
High-dose chemotherapy	3.17 (1.44; 6.99)	0.003	0.37 (0.18; 0.77)	0.008
Tobacco-user (*former or current exposition*)	1.04 (0.41; 2.65)	0.58	1.10 (0.55; 2.18)	0.80
Performance status (≥1 vs. 0)	2.24 (1.04; 4.82)	0.03	1.24 (0.67; 2.30)	0.49
Baseline eGFR (per ml/min/1.73 m^2^ increase)	0.98 (0.96; 0.99)	0.004	0.99 (0.98; 1.01)	0.35
Baseline serum albumin level (per g/dl increase)	0.98 (0.92; 1.04)	0.18	-	-

eGFR: estimated glomerular filtration rate (CKD-EPI formula).

**Table 3 nutrients-16-00889-t003:** Summary of main outcomes according to the level of daily caffeine consumption.

	Overall N = 108	Daily Caffeine Consumption (</≥386 mg/Day)		
		Moderate N = 54	High N = 54	*p*-Value *
Chemotherapy withdrawal				0.396
* AKI*	13 (12.0%)	7 (13.0%)	6 (11.1%)	
* Death*	8 (7.4%)	4 (7.4%)	4 (7.4%)	
* Other*	21 (19.4%)	7 (13.0%)	14 (25.9%)	
Chemotherapy response				0.568
* Progression*	60 (55.6%)	28 (51.9%)	32 (59.3%)	
* Partial response*	31 (28.7%)	18 (33.3%)	13 (24.1%)	
* Complete response*	17 (15.7%)	8 (14.8%)	9 (16.7%)	
Digestive toxicity	22 (20.4%)	6 (11.1%)	16 (29.6%)	**0.032**
Ototoxicity	4 (3.7%)	3 (5.6%)	1 (1.9%)	0.618
Neurotoxicity	13 (12.0%)	5 (9.3%)	8 (14.8%)	0.554
Hematotoxicity	26 (24.1%)	13 (24.1%)	13 (24.1%)	1.000
Death	47 (43.5%)	20 (42.6%)	27 (57.4%)	0.487
Cause of death				0.470
* Progression*	36 (76.6%)	15 (75.0%)	21 (77.8%)	
* Sepsis*	6 (12.8%)	2 (10.0%)	4 (14.8%)	
* Other*/*unknown*	5 (10.6%)	3 (15.0%)	2 (7.4%)	

* comparison with log rank test.

## Data Availability

The raw data supporting the conclusions of this article will be made available by the corresponding author upon reasonable request.

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
