# Peer review of "Daily Caffeine Consumption May Increase the Risk of Acute Kidney Injury Related to Platinum-Salt Chemotherapy in Thoracic Cancer Patients: A Translational Study"

_nutrients, 2024, doi:10.3390/nu16060889_

Round 1

Reviewer 1 Report

Comments and Suggestions for Authors

In this study, lung cancer patients in the original hospital were treated with platinum and the frequencies of acute kidney injury and death were examined. Self-administered questionnaire for caffeine intake over the past 10 years before starting platinum was examined and later the hazard ratio for acute kidney injury was increased to 2.19 in the high caffeine group compared to the low group, but there were no significant differences in mortality. Furthermore, cisplatin and increasing caffeine administration to proximal tubular cells and mice caused increased levels of the renal injury marker NGAL mRNA and blood BUN levels. These findings suggested that lung cancer patients treated with platinum may develop acute kidney injury if they use to drink high dose caffeine.

1)    Lung cancer should be mentioned in the title and abstract.  

2)    The following terms should be explained in the abstract: platinum (cisplatin or carboplatin). Carboplatin should be added to Keywords.

3)    The meaning of the black graph in Figure 2 is not clear. It appears that the mortality rate in the high caffeine group is reduced to about 1/3 compared to the low group. However, Table 2 shows similar mortality HR 1.11 between groups.

4)    Figure 1 and Table 1 look very differently. In Figure 1, the number of patients is larger in the high caffein group than in the low group, but they are identical in Table 1. The way of the low/normal group for caffeine is confusing and should be rewritten as the low group (or low/zero group).

5)    Description should be confirmed in Supplementary Table.

250 à 350 ^ 250-350          26,6 ^ 26.6

Reviewer 2 Report

Comments and Suggestions for Authors this study suggests a potentially deleterious effect of high doses of daily caffeine consumption on the risk of cisplatin-related AKI,
both in clinical and experimental settings.

however, the cut-off of high consumption need to be defined. 

second, the NGAL need to be clarified, why using this marker, as not even mentioned in the introduction part.

Round 2

Reviewer 1 Report

Comments and Suggestions for Authors

Well improved.

Reviewer 2 Report

Comments and Suggestions for Authors

ok